# A Vector-Based Computational Model of Multimodal Insect Learning Walks

**DOI:** 10.3390/biomimetics10110736

**Published:** 2025-11-03

**Authors:** Zhehong Xiang, Xuelong Sun, Jigen Peng

**Affiliations:** 1School of Mathematics and Information Science, Guangzhou University, Guangzhou 510006, China; xiangzhehong@e.gzhu.edu.cn (Z.X.); jgpeng@gzhu.edu.cn (J.P.); 2Machine Life and Intelligence Research Center, Guangzhou University, Guangzhou 510006, China

**Keywords:** ant navigation, learning walk, multisensory model, computational modelling, visual learning, biologically inspired model

## Abstract

Navigation is crucial for animal survival, and despite their small brains, insects are impressive at it. For example, desert ants acquire environmental information by learning to walk before foraging, enabling them to return home accurately over long distances. These learning walks involve multimodal sensory experiences and induce neuroplastic changes in the Central Complex (CX) and the Mushroom Body (MB) of ants’ brains, making them a key topic in behavioural science, neuroscience, and computational modelling. To address unresolved questions in how ants integrate sensory cues and adapt navigation strategies, we propose a computational model that achieves multisensory integration during learning walks. Central to this model is a novel Learning Vector mechanism that dynamically combines visual, olfactory, and path integration inputs to guide movement decisions. Specifically, the agent in our model determines the degree to which it deviates from the homing direction by evaluating the familiarity of the environment. In this way, agents could strike a balance between their tendency to explore and the need to return safely to the nest. Our model replicates key features reported in biological studies and accounts for individual and inter-species variability by tuning parameters such as cue preferences and environmental parameters. This flexibility enables the simulation of species-specific learning walks and supports a unified view of sensory integration and behavioural adaptation. Moreover, it yields testable predictions that may inform future investigations into the neural and behavioural mechanisms underlying insects’ learning walks. How the proposed model could be adapted for robotics navigation has also been discussed.

## 1. Introduction

Navigational ability is essential for animal survival, facilitating crucial tasks such as food foraging, migration, and shelter seeking [1,2,3]. Among animals, insects stand out for their ability to navigate complex environments using brains that are orders of magnitude smaller than those of vertebrates [4,5]. For example, desert ants (*Cataglyphis fortis*) can learn their surroundings within a few days, enabling them to return to their inconspicuous nest entrance from distances of up to several hundred meters [6,7,8]. This remarkable navigational ability has made ants a key subject of study, drawing significant attention from both neuroscience and ethology researchers [9,10,11,12,13,14]. Ethologists [8,15] have found that before ants can demonstrate their impressive navigational skills, they first engage in a learning process. This process involves several preforaging learning walks, during which ants meander in small loops around their nest entrance to familiarize themselves with the surrounding panorama [15,16,17,18,19]. Building upon these findings, neuroscientists [20] have discovered that these small exploratory walks significantly impact the ants’ neural circuits. Specifically, ants that successfully complete learning walks show notable increases in the number of microglomerular synaptic complexes within the mushroom body (MB) collar (MB-co-MG), in the volume of the central complex (CX-vol), and in the number of microglomeruli in the bulb (BU-MG), compared to ants that do not [14,20]. Previous studies have indicated that successful homing (especially visual homing) requires not only the performance of learning walks [6,8] but also that these walks are carried out without spatial restriction [21]. These findings suggest that learning walks play a importance role in establishing ants’ visual homing ability [7,9] and provide a valuable paradigm for studying insect learning [14].

Due to its important role, the learning walk has garnered attention across various fields, including behavioural science [8,16,22], neuroscience [14,20], and computational modelling [23,24]. Behavioural scientists have focused on collecting and analysing trajectory data from various ant species during their learning walks, providing valuable insights into the characteristics of real ants’ learning walks patterns, including empirical data on how trajectory coverage and maximum distance change with the number of learning walks [22], as well as differences in trajectory tortuosity among different species [17]. Across successive learning walks, the exploration area expanded progressively, accompanied by the stepwise acquisition of the nest-centred visual panorama [8,17,22]. In neuroscience, research has revealed that learning walks involve neuroplastic changes [14,25,26] in the central complex (CX) and the mushroom body (MB), which have been shown to be heavily involved in multisensory-driven navigation tasks [27,28,29]. These findings suggest that successful learning walks require specific environmental conditions, such as polarized light, magnetic fields, visual and olfactory cues, which assist ants in navigation. In computational modelling, common navigational tools used by ants—path integration, visual navigation, and olfactory cues—have been incorporated into models constrained by biological principles [23,24,27]. These models simulate the biologically grounded transformation from ambient sensory inputs to navigational decisions, producing strategies that closely align with the behavioural patterns observed in real ants.

Despite that learning walk has been studied from various perspectives across different fields, several key aspects remain unclear. Previous research has discovered that learning walk is a multimodal process, but how these sensory modalities are combined for learning walk remains unclear [9,14,26]. In spite of the hypothesis that different navigation strategies will transit during learning walks [9], it is still unclear how ants decide to change their navigation strategy based on external and internal states and the underlying computational mechanism. Furthermore, the varieties observed between individuals and between different species present additional challenges in revealing the underlying mechanisms [17,22].

To address these gaps, we propose a multisensory model that dynamically balances exploration and homing during ant learning walks, resulting in a unified representation termed the Learning Vector. This novel vector-based modelling approach is inspired by vector computations commonly used in navigation models [23,27,30,31], and is grounded in neural structures found in the insect brain [2,32,33,34]. The Learning Vector provides an intuitive and biologically plausible representation of how ants assess environmental familiarity and make dynamic navigation decisions. In line with previous studies, such vector operations can also be viewed as a simplified implementation of Bayesian-like cue integration, with vector length representing certainty, and thus offer a parsimonious neural framework for modelling the strategy shifts observed during learning walks [35,36]. Specifically, it integrates visual learning via the Mushroom Body (MB) network, innate olfactory cues via the Lateral Horn (LH; simplified here as chemotaxis to enable fast integration of sensory cues, but readily adaptable to support learned olfactory guidance), and path integration via the Central Complex (CX) [27,37,38,39,40,41]. The direction of the Learning Vector determines the agent’s movement at each step, such that exploration during learning walks emerges from deviations away from the homing direction, with the magnitude of deviation increasing when the perceived risk of homing failure is higher. By successfully replicating main characteristics of real ants’ learning walks observed in ethological studies [8,17,22], our model offers a unified explanation for sensory integration and strategy transitions. We found that both individual and interspecies variations in learning walk patterns can be captured by adjusting the intrinsic properties of the model; for instance, altering the visual sensing preference leads to corresponding changes in trajectory tortuosity, consistent with biological hypotheses that ants in visually rich areas tend to show more winding learning paths, while those in visually poor environments move in relatively straighter trajectories [17,22].

In summary, as the first attempt of building a unified multi-sensory computational model of learning walks, this study could provide feasible accounts for the variety of learning walk trajectory tortuosity across species. In addition, based on this model, a set of testable predictions has been proposed to guide future investigations into learning walks. Moreover, guided by the principle of bio-inspired approaches [42,43,44,45], we further discuss how the computational mechanisms of the proposed model can inform robotics applications.

## 2. Methods

All simulations and network models in this study were implemented in Python 3.11, using external libraries including NumPy, Matplotlib, SciPy and OpenCV (cv2). It is important to note that some components of our framework are adapted from our previous work [23]. Specifically, the implementation of the visual environment (Section 2.1.1) and the visual learning module (Section 2.2.1) largely follow this earlier study in order to maintain comparability and leverage validated approaches. By contrast, the key contributions of the present manuscript lie in the introduction of the Learning Vector as a novel mechanism to balance exploratory boldness and caution, the design of a dynamic weighting scheme for multisensory integration, and the associated experimental analyses. These new components go beyond the scope of the previous model and form the central advances of the current work.

### 2.1. Environmental Simulation

To model the sensory environment of ants during exploration, we used a simple yet effective 3D simulation environment that integrates visual and olfactory cues. This environment is sufficient to implement and verify the proposed navigation strategies. A more detailed discussion on the rationale behind this environmental choice and potential improvements can be found in Section 4.

#### 2.1.1. Visual Environment

The visual environment is represented by a 3D model based on the approach of [24].

This model consists of a triangular mesh structure represented by a matrix of size $N_{p}\times 3\times 3$, where each entry defines the 3D coordinates (x,y,z) of the vertices of $N_{p}$ triangular patches (Figure 1A). We note that the environment is the same as that used in our previous work [23], in order to ensure comparability and to build upon a validated framework. The model provides visually perceivable cues, simulating the visual input that ants rely on during navigation. Throughout this study, all distances are measured in meters.

#### 2.1.2. Olfactory Field

To simulate olfactory cues, a Gaussian concentration field was established around the nest, located at the centre of the environment (0,0). The concentration field represents the diffusion of carbon dioxide, modelled as a single point source at the nest. The concentration C(x,y) at any point (x,y) is defined as:(1)$$C\left(x,y\right)=C_{0}\exp\left(-\frac{x^{2}+y^{2}}{2\sigma^{2}}\right),$$
where $C_{0}=1$ denotes the normalized concentration at the nest, and σ controls the diffusion spread. As direct quantitative estimates of olfactory sensitivity during the early stage of learning walks are not yet available, we referred to behavioural studies showing that experienced ants can use nest-associated odour cues for homing within approximately 1 m of the nest entrance [46,47]. In light of this evidence, we adopted a more conservative parameter choice such that agents can effectively rely on olfactory cues only within the initial range of their first learning walks (approximately <0.24 m [22]). This ensures that olfaction is available during early walks, while preventing excessive reliance in subsequent walks. Agent movements are confined to the *x*-*y* plane to simplify the navigation problem.

Based on the home-range marking mechanism of ants [46] and the fact that we only require a submodule that captures the essential characteristics of olfactory-based decision-making. This simplified Gaussian model is sufficient for examining the integration of multimodal cues during learning walks. However, more realistic implementations are possible. For instance, in the presence of wind, an anisotropic Gaussian plume model [31] that incorporates wind direction and velocity into the diffusion process can be employed to simulate a more biologically plausible odour field (the related experiments can be found in the Appendix A). Moreover, olfactory learning mechanisms could be introduced by incorporating a Mushroom Body-like network for odour representation, analogous to our visual learning module. These extensions are straightforward to implement and fully compatible with our current framework; however, in this study we focus on how Learning Vectors integrate multiple sensory cues. For further discussion on the rationale and extensibility of our environmental model, see Section 4.

### 2.2. Navigation Models

To enable biologically inspired simulation of ant exploration, we adapted a navigation model grounded in previous work. This model integrates three key navigation strategies—path integration [27,30], visual learning [23,48], and innate olfactory navigation [31,41]—each implemented based on behavioural principles observed in natural ant systems. By combining these strategies, the model reproduces complex exploratory behaviours and supports the investigation of sensory integration during navigation.

We model the neurons in the proposed network using a simple firing rate mechanism. Unless otherwise specified, the output firing rate *C* is represented as a sigmoid function of the input *I*, defined by:(2)$$C=\frac{1}{1+e^{-I}}.$$

#### 2.2.1. Visual Learning

Previous studies have demonstrated that ants are capable of employing visual memories to navigate extended routes through complex environments. During learning walks, panoramic views in the vicinity of the nest are acquired and encoded within the mushroom body as long-term visual memories. Subsequent homing can then be achieved by iteratively comparing the currently perceived view with the stored representations, allowing the ant to progressively align its trajectory with familiar locations and thereby accomplish visual homing [48]. The visual learning process is modelled using a simplified mushroom body (MB) neural network, designed to capture the core principles of visual memory in insects [20,37,49]. Visual input is encoded using Zernike Moments (ZM) [50,51], a rotationally invariant feature representation derived from preprocessed panoramic snapshots. This choice allows the model to construct a stable visual familiarity field centred around the nest, independent of the agent’s viewing direction [31].

The MB network follows the anatomical structure observed in insects, consisting of three layers: Visual Projection Neurons (VPNs), Kenyon Cells (KCs), and Mushroom Body Output Neurons (MBONs). The MB-network evaluates image familiarity through a three-layer architecture. The VPN layer receives visual features (e.g., raw pixel intensities or Zernike Moment descriptors). Each KC neuron is randomly connected to a subset of VPN neurons, forming a sparse representation, while the MBON neuron receives weighted inputs from all KCs. At initialization, all MBON weights are identical, and its response corresponds to the weighted sum of KC activity. During training, image features activate the corresponding VPNs, thereby driving connected KCs. According to an associative learning rule inspired by Kenyon cell plasticity, activation of a KC leads to a reduction of its synaptic weight onto the MBON. When similar images are encountered again, overlapping features preferentially activate KCs with weakened weights, resulting in a lower MBON output. Thus, the MBON response provides a scalar measure of image novelty. This part of the work is not original to the present study, and further details of the MB-network can be found in previous work [23,48].

In this model, we assume that visual learning primarily occurs at scanning events, without modelling potential latent learning that may take place during intermediate movement steps. This simplification allows us to focus on scan-triggered learning, which plays a key role during early learning walks [52]. Details regarding when and how scanning events occur during simulations can be found in Section 2.4.

At any position (x,y), the agent’s heading θ is defined in an allocentric coordinate system relative to the nest. During scanning, for each heading θ, a visual snapshot is captured at the neighbouring location (x+Ucosθ,y+Usinθ), where *U* denotes the step size. The Zernike Moment amplitudes $A_{i}\left(\theta \right)$ (considered as image features and provided as input to the MB network; the detailed computation of ZM can be found in Appendix A) at this new position are processed by the trained mushroom body (MB) network to compute the novelty score $C_{\mathrm{MBON}}\left(x+U\cos\theta ,\;y+U\sin\theta \right)$. The novelty score at the current position, $C_{\mathrm{MBON}}\left(x,y\right)$, is calculated in the same manner.

To obtain a relative measure of visual familiarity, the novelty score is normalized as follows:(3)$$\omega_{v}\left(x,y\right)=1-\frac{C_{\mathrm{MBON}}\left(x,y\right)}{N_{\mathrm{KC}}},$$
where $N_{\mathrm{KC}}$ is the total number of Kenyon Cells in the MB network. The resulting score $\omega_{v}\left(x,y\right)$ ranges from 0 (completely unfamiliar) to 1 (highly familiar), with higher values indicating greater familiarity at the given location.

To estimate the direction of the nest, the agent evaluates how the novelty score changes in different directions. Since lower values of $C_{\mathrm{MBON}}$ indicate more familiar scenes, the agent selects the heading $\theta_{v}$ that maximizes the decrease in novelty:(4)$$\theta_{v}=\underset{\theta }{arg max}\left(C_{\mathrm{MBON}}\left(x,y\right)-C_{\mathrm{MBON}}\left(x+U\cos\theta ,\;y+U\sin\theta \right)\right),$$

This computation identifies the direction along which the visual novelty decreases most steeply, guiding the agent toward the most familiar nearby region.

#### 2.2.2. Olfactory Navigation

As in many other animals, ants are also able to utilise odour cues to support navigation. In the context of homing, they employ a home-range marking mechanism, whereby colony-specific chemical blends are actively or passively deposited in the immediate vicinity of the nest entrance. By following the gradient of these chemical cues, ants can locate the nest through an innate odour-gradient tracking mechanism [9,41,46]. In our model, innate olfactory navigation was incorporated as the primary submodule for integration.

At any position (x,y), the agent perceives the olfactory concentration C(x,y) from the Gaussian field. Since the concentration has been normalized in the environment setup, the familiarity score can be directly defined as: (5)$$\omega_{\mathrm{olf}}\left(x,y\right)=C\left(x,y\right).$$

To determine the homing direction, the agent computes the local gradient of C(x,y) by directionally sampling the olfactory concentration around the current position.(6)$$\nabla C\left(x,y\right)=\frac{1}{\sigma^{2}}\exp\left(-\frac{x^{2}+y^{2}}{2\sigma^{2}}\right)\cdot \left(-x,-y\right).$$

The direction of the steepest gradient ascent, denoted as $\theta_{\mathrm{olf}}$, is determined using the atan2(y,x) function, defined as:(7)$$\mathrm{atan}2\left(y,x\right)=\left\{\begin{array}{cc} \arctan\left(\frac{y}{x}\right) & x>0,\\ \arctan\left(\frac{y}{x}\right)+\pi & y\geq 0,\;x<0,\\ \arctan\left(\frac{y}{x}\right)-\pi & y<0,\;x<0,\\ +\frac{\pi }{2} & y>0,\;x=0,\\ -\frac{\pi }{2} & y<0,\;x=0,\\ \mathrm{undefined} & y=0,\;x=0. \end{array}\right.$$

The homing direction is then given by:(8)$$\theta_{\mathrm{olf}}=\mathrm{atan}2\left(-y,-x\right).$$

This strategy simulates the agents’ gradient ascent process, generating an effective olfactory decision vector for subsequent cue integration.

#### 2.2.3. Path Integration

Path integration (PI) is a fundamental navigation mechanism observed in many insects, including ants and bees [24,27]. It allows an agent to estimate its current position relative to the starting point (e.g., the nest) by integrating self-motion cues such as direction and distance travelled over time. This process is believed to rely on specific neural circuits in the central complex (CX) of the insect brain, as suggested by Stone et al. [27]. The neural model proposed in this study provides a biologically constrained framework for understanding how path integration is achieved and processed. In order to focus on the integration of submodules leading to the Learning Vector, we simplified the neural implementation of path integration and employed a more basic vector accumulation approach to enable this submodule to perform the function of path integration. Further possibilities for optimising each submodule are discussed in the Section 4.

In our simulation, the homing vector $V_{\mathrm{PI}}$ is computed by accumulating movement vectors v(t), which represent the agent’s direction and travelled distance at each time step *t*. To account for biological constraints, small inaccuracies are introduced at each step. The homing vector is therefore expressed as:(9)$$v_{\mathrm{PI}}=\sum_{t=0}^{T}\left[v(t)+\delta (t)\right],$$
where v(t) is the intended movement vector, and δ(t) represents small errors caused by biological noise or inaccuracies in estimating movement (approximated as 1% of the step length per direction, i.e., following a normal distribution N(0,0.01cm)).

The resultant homing direction indicated by the PI mechanism, denoted as $\theta_{\mathrm{PI}}$, is obtained by computing the angular direction (phase) of the resultant homing vector $V_{\mathrm{PI}}$:(10)$$\theta_{\mathrm{PI}}=\mathrm{atan}2\left(V_{\mathrm{PI},y},V_{\mathrm{PI},x}\right),$$
where $V_{\mathrm{PI},x}$ and $V_{\mathrm{PI},y}$ are the *x*- and *y*-components of $V_{\mathrm{PI}}$, respectively.

This formulation reflects the inherent noise and limitations in biological systems, where small inaccuracies δ(t) accumulate over time, leading to gradual deviations in the homing vector.

### 2.3. Leaning Vector

While individual navigation strategy such as vision, olfaction, and path integration have been extensively studied in ant navigation, existing research lacks an integrated computational model that simulates how these cues are dynamically combined during learning walks. This gap limits our understanding of how behavioural strategies emerge from multisensory processing. In this section, we propose a biologically inspired model that bridges behavioural findings, neural mechanisms, and computational modelling. Our approach introduces a unified framework that combines multiple navigation tools through dynamic weighting, enabling real-time estimation of both homing direction and exploration strategy. We first define the Homing Vector, which integrates directional cues from each modality. We then extend this to the Learning Vector, which modulates the homing signal based on environmental familiarity and internal learning state, ultimately generating trajectories that reflect adaptive ant navigation behaviour in ants’ learning walks.

Ant navigation during learning walks relies on multiple sensory modalities to estimate the direction. At each step, the agent evaluates this direction based on three core navigation tools: visual familiarity, olfactory concentration, and path integration. Each tool provides an angular estimate of the nest direction, denoted as $\theta_{v}$, $\theta_{\mathrm{olf}}$, and $\theta_{\mathrm{PI}}$, respectively.

To model the reliability of each strategy cue, we assign normalized familiarity scores—$\omega_{v}$ for vision and $\omega_{\mathrm{olf}}$ for olfaction—ranging from 0 (unfamiliar) to 1 (highly familiar). These scores reflect the confidence in the corresponding directional estimate at a given location.

From the ethnological perspective, while PI is continuously available, its contribution varies with the availability of other sensory inputs. Specifically, ants tend to rely more heavily on PI in unfamiliar environments, and shift toward vision-based navigation as environmental familiarity increases [7,9,21]. Based on these findings, we hypothesize that path integration serves as a fallback strategy when visual and olfactory cues are weak, enabling adaptive compensation in unfamiliar environments. This behavioural evidence directly motivated the weighting scheme adopted in our model. In our model, we define the PI weight as:(11)$$\omega_{\mathrm{PI}}=\max\left\{1-\omega_{v}-\omega_{\mathrm{olf}},0\right\},$$
which ensures non-negative values and allows the PI component to increase adaptively as the availability of other sensory information decreases.It should be noted that the weighting mechanism in this formula constitutes a modelling assumption, proposed to account for the integration of path integration, visual navigation, and innate olfactory navigation within the context of learning walks.

Formally, the Homing Vector $V_{\mathrm{Homing}}$ is computed as the weighted sum of the three directional vectors: (12)$$V_{\mathrm{Homing}}=\omega_{v}\left(\begin{array}{c} \cos\theta_{v}\\ \sin\theta_{v} \end{array}\right)+\omega_{\mathrm{olf}}\left(\begin{array}{c} \cos\theta_{\mathrm{olf}}\\ \sin\theta_{\mathrm{olf}} \end{array}\right)+\omega_{\mathrm{PI}}\left(\begin{array}{c} \cos\theta_{\mathrm{PI}}\\ \sin\theta_{\mathrm{PI}} \end{array}\right).$$

This vector represents the agent’s integrated estimate of the nest direction based on multisensory navigation tools, and serves as the foundational cue for downstream navigation decisions.

The Homing Vector estimates the direction toward the nest based on multisensory input, but by itself it does not fully capture the behavioural trade-offs that ants face during learning walks. In this context, ants are not only concerned with returning to the nest but also need to evaluate whether to continue exploring or stay within familiar areas for safety. This decision depends on both their perceived familiarity with the environment and their spatial relationship to the nest.

To model this process, we introduce the Learning Vector, which modifies the Homing Vector by applying an angular offset ϕ that reflects the agent’s current behavioural priority between exploration and homing. This offset is determined by the degree of “familiarity” across three navigation strategies: vision, olfaction, and path integration. Each strategy contributes a monotonic mapping from input to offset magnitude—higher “familiarity” encourages exploration, resulting in a larger ϕ.

In addition, to reflect the temporal structure of learning walks—where exploration eventually concludes—we define an internal state variable $A_{c}$. When exploration is ongoing, $A_{c}=1$, and the offset is computed normally. Once a predefined exploration step limit is reached, $A_{c}=0$, and the agent switches to full homing behaviour.

This design allows the model to dynamically balance exploration and safety based on both real-time environmental input and internal learning progress. The Learning Vector thus serves as a biologically plausible and computationally tractable mechanism for simulating ant behaviour during learning walks.

To compute the Learning Vector, we first calculate an angular offset ϕ that reflects the behavioural trade-off between homing and exploration. The offset is derived from the weighted sum of three modality-specific strategy mapping functions: $F_{v}\left(\cdot \right)$, $F_{\mathrm{olf}}\left(\cdot \right)$, and $F_{\mathrm{PI}}\left(\cdot \right)$, each of which maps a normalized familiarity score to an angular deviation in the range [0,π]. These functions are designed to be monotonic: as familiarity increases, the corresponding angular offset also increases, promoting greater exploration.

In our model, we construct the following strategy-specific familiarity mapping functions. For the visual familiarity function, the agent maintains a sequence $V_{n}$ that records the MB network output at each step *n*. Within a sliding window of size $N_{\mathrm{window}}$, the agent computes the mean ${\bar{V}}_{n}$ and the standard deviation $\sigma_{V_{n}}$ over the values $\left\{V_{n-k+1},\ldots ,V_{n}\right\}$. We then define the lower and upper thresholds as:$$m_{V_{n}}={\bar{V}}_{n}-\sigma_{V_{n}},\;M_{V_{n}}={\bar{V}}_{n}+\sigma_{V_{n}}.$$

Since lower MBON scores correspond to higher familiarity, the visual familiarity function $F_{v}\left(V_{n}\right)$ is defined as a piecewise linear mapping passing through the points $\left(m_{V_{n}},\pi \right)$ and $\left(M_{V_{n}},0\right)$. Formally,(13)$$F_{v}\left(V_{n}\right)=\left\{\begin{array}{cc} \pi , & V_{n}\leq m_{V_{n}},\\ \frac{M_{V_{n}}-V_{n}}{M_{V_{n}}-m_{V_{n}}}\cdot \pi , & m_{V_{n}}<V_{n}<M_{V_{n}},\\ 0, & V_{n}\geq M_{V_{n}}. \end{array}\right.$$

For the olfactory familiarity function, we directly map the relative concentration at the agent’s current position, C(x,y)∈[0,1], to an angular deviation in the range [0,π]. Formally,(14)$$F_{\mathrm{olf}}\left(C\left(x,y\right)\right)=C\left(x,y\right)\cdot \pi .$$

For the familiarity of PI, we introduce a parameter $d_{1}$, representing the maximum distance reached during the first learning walk. In the subsequent learning walks, this distance is increased according to the multiplicative factors reported in Deeti’s study [22]. At the *k*-th learning walk, the familiarity of the PI strategy can then be quantified by taking the ratio between the current path integration vector length *r* and $d_{k}$: (15)$$F_{\mathrm{PI},k}\left(r\right)=\frac{r}{d_{k}},$$
where *r* denotes the current path integration vector length and $d_{k}$ is the expected maximum distance at the *k*-th learning walk. By combining the three strategy-specific familiarity functions, the overall angular offset ϕ is obtained as:(16)$$\phi =A_{c}\cdot \left(\omega_{v}\cdot F_{v}\left(\omega_{v}\right)+\omega_{\mathrm{olf}}\cdot F_{\mathrm{olf}}\left(\omega_{\mathrm{olf}}\right)+\omega_{\mathrm{PI}}\cdot F_{\mathrm{PI}}\left(r\right)\right)$$

Here, $\omega_{v}$, $\omega_{\mathrm{olf}}$, and $\omega_{\mathrm{PI}}$ are the normalized familiarity values for vision, olfaction, and path integration, respectively, and *r* is the length of the homing vector derived from the path integration system. The offset ϕ is modulated by a state variable $A_{c}$, which governs whether the agent continues exploring ($A_{c}=1$) or switches to homing mode ($A_{c}=0$) once a predetermined number of steps $n_{p}$ is reached.

We define the Learning Vector by rotating the Homing Vector by ϕ using a standard 2D rotation matrix:(17)$$V_{\mathrm{Learning}}=\left(\begin{array}{cc} \cos\phi & -\sin\phi \\ \sin\phi & \cos\phi \end{array}\right)V_{\mathrm{Homing}}.$$

This vector encodes the direction that balances exploration and homing according to current sensory inputs and internal state. Through this transformation, the Learning Vector enables real-time navigation decisions that flexibly adapt to environmental familiarity and task demands.

In the following subsection, we describe how the Learning Vector is used iteratively to guide the agent’s learning walk across spatial environments in the simulations.

### 2.4. Simulation and Validation

#### 2.4.1. Agent

Based on empirical data from [22], which show that learning walk durations increase exponentially across the first four trials, we assume a constant walking speed and model the expected number of movement steps in each learning walk *j* as:(18)$$n_{pj}=n_{p1}\cdot r^{j-1},$$
where $n_{p1}$ is the expected number of steps in the first walk and r=2 is the fixed rate of growth.

Additionally, the average number of scans $n_{j}$ observed in real ants during each walk is extracted from the same empirical study. Based on these two quantities, we define the scanning interval for each learning walk as:(19)$$I_{\mathrm{scan}}=\frac{n_{pj}}{n_{j}},$$
which yields the average interval of steps between consecutive scans. This approach ensures that the agent performs a biologically plausible number of scans per walk, matching empirical data while avoiding unrealistically frequent scanning in longer walks.

To simulate the learning walk behaviour of ants, we define the model as L, which integrates multisensory inputs and outputs the Learning Vector $V_{\mathrm{Learning}}^{n}$ at each time step *n*. Specifically, the model receives the agent’s current spatial position $\left(x_{n},y_{n}\right)$, the corresponding image snapshot $Img_{n}$, the olfactory concentration $C(x_{n},y_{n})$, and the current path integration vector $v_{\mathrm{PI}}^{n}$, and returns a real-time decision vector:(20)$$V_{\mathrm{Learning}}^{n}=L\left(x_{n},y_{n},Img_{n},C\left(x_{n},y_{n}\right),v_{\mathrm{PI}}^{n}\right)$$

The agent is initialized by constructing the basic architecture of the MB network, including the random connectivity between VPNs and KCs, together with an initial position $\left(x_{0},y_{0}\right)$, a heading direction, and a cumulative path integration vector. It then updates its position at each step by moving in the direction specified by the Learning Vector. Here, $\left(x_{n},y_{n}\right)$ denotes the simulated agent’s coordinates in the environment for the purpose of model implementation and trajectory generation. In the biological counterpart, the agent does not require explicit global localization to obtain these inputs; rather, each strategy module (visual, olfactory, PI) operates on locally available information, and the homing distance *r* can be inferred from the PI system. The coordinate notation is introduced here solely to facilitate a concise formulation of the iterative update in the simulation framework. The position at step *n* is computed iteratively as:(21)$$\left(x_{n},y_{n}\right)=\left(x_{n-1},y_{n-1}\right)+U\cdot \frac{V_{\mathrm{Learning}}^{n-1}}{∥V_{\mathrm{Learning}}^{n-1}∥},$$
where *U* is the agent’s fixed step size. This formulation ensures that the direction of motion dynamically adapts to the integrated multisensory input and internal learning state at each time point.

By iteratively applying Equations (Equation 20) and (Equation 21), the model generates full learning walk trajectories that reflect the ant’s exploration strategy and progression over time.

To further mimic the natural variability observed during exploratory behaviours, the agent occasionally performs stochastic steps rather than strictly following the computed Learning Vector. Specifically, at each time step, with a probability $p_{\mathrm{rand}}$, the agent samples a random movement direction from a von Mises distribution centred at its current heading (μ=0) with a concentration parameter κ ranging from 5 to 50. The agent then moves along the sampled random direction, while maintaining a fixed step size *U*. Otherwise, it follows the direction specified by the Learning Vector. This mechanism allows the agent to occasionally deviate from deterministic strategies, promoting exploration of novel areas and reducing trajectory redundancy.

All simulation parameters used in this study are summarized in Table 1 for reference in the subsequent Results section. For the MB network, parameters such as the visual learning rate $\lambda_{\mathrm{vis}}$, the Kenyon Cell activation threshold $\theta_{\mathrm{KC}}$, the number of Visual Projection Neurons $N_{\mathrm{VPN}}$, and the number of Kenyon Cells $N_{\mathrm{KC}}$ were designed in reference to previous models of the mushroom body network [31,48]. The spread of the Gaussian odour field σ was chosen as a conservative estimate, reflecting the olfactory navigation ability of ants [41,53]. The familiarity-to-offset mapping functions $F_{v},F_{\mathrm{olf}},F_{\mathrm{PI}}$ and the scanning interval $I_{\mathrm{scan}}$ were defined by Equations (Equation 13), (Equation 14), (Equation 15) and (Equation 19) in the main text. The remaining parameters, including the step scaling factor *r*, the random step probability $p_{\mathrm{rand}}$, the step size *U*, and the von Mises concentration parameter κ, are directly related to Learning Walk behaviours. By dynamically adjusting these parameters, our model is able to reproduce different types of Learning Walk trajectories.

#### 2.4.2. Validation

To evaluate the outcomes of the learning walks generated by our model—given that the main objective of insect learning can be considered to involve successful homing—we assessed the homing performance of agents after learning in a series of simulated experiments under controlled conditions (no visual learning occurs during the testing process). Ten agents with varying levels of learning experience were tested from 64 distinct release points, arranged at eight evenly spaced distances (1 to 8 m) and eight angular directions (0°, 45°, …, 315°) surrounding the nest. The choice of tenagents was made to align with the sample sizes commonly used in behavioural studies [22], thereby ensuring both biological plausibility and statistical consistency.

During the testing phase, learning was disabled to simulate a memory recall scenario, ensuring that agents relied solely on previously acquired visual information from learning walks, in combination with their innate olfactory cues, without further updating their internal representations. For each agent, its ability to return to the nest from each release point was recorded, and the success rate was calculated as the proportion of trials in which the agent reached the nest successfully. This design enabled the assessment of navigational performance across both short-range (1–4 m) and long-range (5–8 m) conditions.

To quantify the effect of learning experience, we compared homing success rates between agents that had undergone different numbers of learning walks. Before performing pairwise comparisons, we first examined whether the data satisfied the normality assumption. For data that met this criterion, we used paired *t*-tests (Table 2) and reported effect sizes using Cohen’s *d* (Table 3). For data that violated the normality assumption, we applied Wilcoxon signed-rank tests and reported effect sizes as *r* values. The full statistical and effect size results for close-, middle-, and far-range comparisons are provided in the Appendix A.

#### 2.4.3. Generalizability

To evaluate the model’s generalisation capacity under altered sensory conditions, we implemented a variant of the agent with reduced reliance on visual input. In this condition, all model components and parameters were kept identical to the baseline setting, except for two targeted modifications: (1) the visual learning rate $\lambda_{\mathrm{vis}}$ was reduced from 0.01 to 0.005, and (2) the pirouette scanning frequency was set to one quarter of the default rate. These changes were designed to simulate less vision-reliant individuals, while maintaining the same multiplicative increase in scan counts across learning walks as reported in [22]. All other strategy weights, memory structures, and network configurations remained unchanged, allowing for a direct comparison with the standard agent configuration.

## 3. Results

In this section, we first show how the proposed model could replicate the biological data on maximum trajectory distance and coverage area [22] under similar experiment settings with a similarly rich visual environment. Then, the effectiveness of the visual learning driven by our learning walks model is evaluated by the agent’s solving visual homing task [54]. To explore this further, we examine whether our adaptive weighting mechanism can reproduce the transition of navigational strategies hypothesised in behavioural studies [9,21]. Finally, we found that by adjusting the parameters of the ‘pirouettes’, our model also provides a potential account for species-specific behaviours [17].

### 3.1. Replicate the Characteristics of Real Ant’s Learning Walk

According to ethological studies of learning walks, trajectories analysis is often regarded as the primary approach to reveal the mechanisms underlying the ant’s learning walks. To align with this, we first generate the agent’s trajectories guided by the proposed learning walk model and then analysis them in a identical way as that in biological studies. The results are shown in Figure 2A–C, which presents four successive learning walks of three individual agents. Figure 2D shows the trajectories of the second learning walk of all agents. Consistent with real biological data, the ants progressively expanded their coverage with each additional walk: their exploration was initially concentrated near the nest and gradually extended outward [6,7,8].

Biological studies have reported that while some ants tend to restrict their exploration to a particular direction, others exhibit more distributed, multi-directional patterns [22]. This phenomenon was also observed in our models, see Figure 2A for learning walks exhibiting preferred exploration direction, and see Figure 2B,C for individuals performing broader range of exploratory direction. Note that we did not intentionally set the exploration direction in the model, this phenomenon may emerge from some random process, indicating that the observation of real ants may also be the same case. However, this requires further investigation, which is beyond the scope of this study.

In ethological studies, two key geometrical metrics of the ant’s learning walk trajectories, the trajectory coverage area and the maximum distance reached, have been identified to quantitatively describe the ’degree of exploration’. Changes in these metrics are believed to reflect the ant’s increasing familiarity with the environment, as exploration gradually expands [22]. Figure 2E plots the agent’s distance to nest during the second learning walk of each individual agent (trajectories are plotted in Figure 2D), the maximum distance reached of our simulation and the biological experiments are marked. From which we could see that the results of the proposed model largely overlapped with the data of real ants [22]. For further comparison of maximum distance reached in other times (i.e., the first, third, and fourth) of learning walks, see the right panel of Figure 3. The comparison of the trajectory coverage area between our computational simulation and biological experiments are shown in the left panel of Figure 3. These results indicate that our results are broadly consistent with the ranges observed in real ants. Across 40 trials, 85% of the simulated area values and 90% of the maximum distance values fell within empirical bounds. For the remaining outliers, the average relative deviation was modest: 1.29% for trajectory coverage area and 0.89% for maximum distance reached. These results provide quantitative support that the model could capture the overall spatial characteristics of real ants’ trajectories. Building on this, the proposed Learning Vector and weighting mechanism could help to explain the variation in maximum distance and coverage observed in real ants’ learning walk trajectories. It is also possible that ants adjust the degree of boldness or conservativeness in their learning walks according to their familiarity with the environment.

### 3.2. Evaluate Learning Performance by Visual Homing

Learning walks allow ants to gradually become familiar with their surroundings and are considered a key process in enabling accurate visual homing [8,9]. To test whether our model supports this fundamental function, we simulated 10 agents, each performing four learning walks, followed by a Homing Test [54]. The purpose of this test is to evaluate whether the agent could acquire useful visual information during exploration guided by our learning walks model and use it for a successful return to the nest. To this end, we adopted the widely used computational model of the mushroom bodies (MBs) [23,48] to simulate visual learning process during the ant’s learning walk. More precisely, the visual scenes encountered throughout the learning walk are used as inputs (i.e., a training dataset) feed into the Mushroom Body network.

We analyse how MBON activity (simulated neural responses, not biological recordings) varied over time during the learning walks (Figure 4B), and how the spatial distribution of familiarity evolved in the learning area (Figure 4C). As shown in Figure 4B, MBON values during homing decreased progressively with each successive learning walk. This trend suggests that the agent became increasingly familiar with visual input encountered during navigation. Figure 4C further illustrates that areas with low MBON activation expanded after each learning session, indicating that the MB network has acquired a broader spatial familiarity.

Inspired by behavioural paradigms commonly used in ant navigation studies [54], we designed a visual homing test to evaluate whether the visual information acquired during learning walks led to improved homing performance. In this test, we measured the number of successful returns to the nest from various test locations after different amounts of learning experience. Figure 4D shows representative homing attempts from four fixed positions after the first to the fourth learning walks. Figure 4E presents the homing success counts for a single agent released near (1–4 m, blue curve, largely within the learning walk area) and far (5–8 m, orange curve, largely outside the learning walk area) from the nest (see details in Section 2), exhibiting an overall increase with each learning session. Figure 4F further visualises the spatial distribution of homing success across all test locations, wherein the black squares indicate successful returns to the nest at that location while the white squares represent failures.

To assess performance at the group level, Figure 5 summarizes the success rates of ten simulated ants after each learning walk. Both near-range (1–4 m) and far-range (5–8 m) success rates increased substantially with experience. Specifically, agents trained with four learning walks significantly outperformed those trained with only one in both near-range trials (*t*-test, t=3.375, $p_{\mathrm{Holm}}=0.0492$, d=1.067) and far-range trials (Wilcoxon signed-rank test, $p_{\mathrm{Holm}}=0.0375$, r=0.886). Statistical analysis across all 64 test points further confirmed a significant improvement in overall homing success rate from the first to the fourth learning walk (*t*-test, t=7.740, $p_{\mathrm{Holm}}=0.00018$, d=2.799). Further statistical results can be found in the Appendix A, where pairedttestresults.xls documents the statistical outcomes for the close-, middle-, and far-range comparisons. In addition, we provide all original test results in Appendix A.

Notably, the homing success rates did not show a statistically significant improvement between the second learning walk and the third learning walk (*t*-test, t=1.121, $p_{\mathrm{holm}}=0.29135$, d=0.553), while the difference between the first learning walk and the fourth learning walk was highly significant. This result suggests that a single learning walk may not consistently lead to an immediate improvement in visual homing ability. However, continued visual exploration through multiple learning walks can reliably enhance navigational performance over time. By comparing the trajectory in Appendix A with the homing success trend in Appendix A, we speculate that the main exploration directions during the second learning walk to the fourth learning walk may have partially diverged from that of the first learning walk. As a result, the visual memories acquired in the first learning walk might not have sufficiently supported homing from the tested locations, leading to a temporary performance drop after the second walk. Nonetheless, as the learning walks progressed, the agent gradually expanded its visual coverage, resulting in steady recovery and eventual improvement in homing success.

In summary, our results suggest that the model may enhance learning performance from both internal perspective, as indicated by a reduction in the simulated MBON values of our neural network model representing increased environmental familiarity, and behavioural perspective as reflected by an improvement in homing ability observed at the behavioural level. However, the extent of improvement varied across learning sessions and was not always statistically significant. This finding appears to suggest that improvements in visual homing ability may not progress in a strictly linear manner with each additional learning walk. Rather, it is possible that the effectiveness of each learning session depends on factors such as the direction of exploration or how well it aligns with previously acquired visual memories.

### 3.3. Adaptive Weighting Could Explain the Navigational Strategy Transition

Learning walks in ants are known to involve multiple sensory modalities [9,14,26,55], including olfactory, visual, and path integration cues. However, the mechanism by which these signals are dynamically integrated during learning walks remains unclear. Behavioural studies have suggested that ants may shift between navigation strategies at different stages of a learning walk, but the sensory context that drive such transitions are not well understood [9,21]. To address this, we implement a cue fusion mechanism. This approach reflects the idea that strategy may contribute differently under varying environmental conditions. Specifically, olfactory weight is modulated by local odour concentration, visual weight is determined by MBON output representing visual familiarity, and path integration serves as a compensatory fallback when other cues are weak. These weights are computed in real time and reflect the agent’s relative reliance on each strategy. By analysing these weights, we aim to quantify how cue usage evolves both within (Figure 6A) and across (Figure 6B) learning walks.

Figure 6A reveals three distinct phases of cue usage during the second learning walk (see Figure 2D). Olfactory cues dominate in the early and late stages (see the x-axis, steps 0–10 and 90–100), while their weight drops below 10% during the middle phase (steps 30–70). Path integration follows the opposite trend, peaking at over 60% in the middle phase and dropping below 10% at the beginning and end. Visual cue weights remain modest throughout, but show a mild upward trend. Notably, cue variability (as indicated by the height of the boxes) remains relatively low within each phase but increases during transitions between phases. This increased variability suggests that individuals diverge more strongly during strategy switching. A possible explanation is that different individuals, based on their prior learning experiences, may form distinct judgements about their familiarity with the environment. Consequently, they adjust the weighting of navigational cues differently, which in turn influences their subsequent learning processes [9,16,22].

Figure 6B presents the average cue weights for each of the four learning walks. In the first learning walk, the agent relied primarily on olfactory (61%) and path integration (31%) cues, with limited visual input (8%), reflecting a typical near-nest strategy [9,22]. In the second walk, olfactory weight declined to 17%, while path integration increased to 66%, becoming the dominant cue as visual familiarity remained limited. In the third and fourth walks, visual weight increased to 30% and 41%, respectively, while olfaction dropped below 7%. Path integration remained high (62% and 52%), resulting in a combined dominance of vision and PI exceeding 90%. The weight dynamics in our model mirror the ant behavioural continuum from PI-dependence (naïve) to vision-dependence (experienced) [21]. These results suggest that cue reweighting, as proposed in behavioural studies, can effectively support the generation of plausible navigation strategies—at least within the context of learning walks.

### 3.4. Account for Species-Specific Behaviour

Biological studies have revealed considerable interspecific variation in learning walk behaviour among ant species [16,17,22,55]. In particular, species differ in their scanning frequency, turning dynamics, and relative reliance on different strategy during navigation. For instance, *Cataglyphis* species inhabiting visually sparse desert salt pans exhibit minimal scanning activity and are thought to depend less on visual cues during the early stages of navigation [17]. In contrast, Melophorus bagoti, which inhabits visually richer environments with relatively more vegetation (e.g., buffel grass (Pennisetum cenchroides), a mosaic of Acacia bushes, and Eucalyptus trees), shows increased scanning activity during learning walks and may rely more on visual cues in the early stages of navigation [22]. These behavioural differences underscore the need for computational models capable of reproducing species-specific traits through biologically grounded parameter adjustments.

To investigate interspecific variability in learning walk behaviour using our model, we simulated agents with reduced visual capabilities inspired by ants from visually impoverished habitats (For more repeated experiments, please refer to the Appendix A). We assumed that agents across environments do not differ markedly in basic motor parameters, such as step size and walk duration, thereby allowing us to isolate the effect of visual reliance. In the simulation, the number of scanning events (pirouettes) was reduced, and the learning rate in the Mushroom Body (MB) network was lowered to mimic slower visual memory formation. These less vision-reliance agents, with all other parameters held constant, performed four learning walks in our simulation. Their trajectories, cue weight dynamics, and behavioural metrics were subsequently compared with those of agents under standard visual conditions.

Figure 7A presents representative trajectories across the four learning walks, revealing a marked reduction in tortuosity for less vision-reliance agents, particularly by the fourth learning walk as the weights of vision increased. To quantify this, we calculated the mean turning angle per step (Figure 7B), which showed a 37.93% decrease compared to the normal-vision group (blue bars), indicating smoother routes. The number of scanning events (Figure 7C) also decreased accordingly, further strengthening the behavioural divergence. Cue weight dynamics (Figure 7D) revealed that olfactory reliance remained largely comparable between groups (less than 5% difference), while visual cue dependence was substantially higher in normal-vision agents (25.71% increase by Walk 4). In contrast, less vision-reliance agents relied more heavily on path integration, with a 29.63% increase in PI weight compared to their normal-vision counterparts.

Collectively, our model demonstrates how targeted parameter adjustments can recapitulate species-typical navigation strategies. Notably, the emergence of smoother trajectories in low-vision agents—achieved solely by reducing MB learning rates and pirouette frequency—can be linked to interspecies differences observed in visually sparse habitats [6,17]. Specifically, decreasing the MB learning rate simulates a population with weaker visual learning capabilities, potentially reflecting long-term adaptation to impoverished visual environments. Reducing pirouette frequency, on the other hand, corresponds to a behavioural trait that limits the number of learning samples available during early experience, consistent with ethological observations [17]. Together, these manipulations allow us to simulate navigation patterns characteristic of such species. This case exemplifies the model’s generalisability and its potential to simulate interspecific behavioural diversity through the tuning of biologically relevant parameters. Such flexibility offers a valuable tool for future studies exploring how ecological context shapes multimodal integration strategies in ant navigation.

## 4. Conclusions and Discussion

This study aimed to explore the multimodal mechanisms underlying learning walk behaviour in ants, as well as the potential integration of these mechanisms during transitions in navigational strategies. To this end, we proposed a multisensory learning walk model that integrates visual, olfactory, and path integration cues. The model successfully reproduced the characteristic trajectory patterns and adaptive strategy shifts observed in ethological studies of ants. Its validity and generalisability were demonstrated through a series of Visual Homing Tests and simulations tailored to reflect species-specific behavioural traits. In summary, our contributions lie mainly in (1) proposing the concept of the Learning Vector, which characterises the balance between exploratory boldness and caution based on environmental familiarity, and (2) hypothesising a special dynamic weighting mechanism that can be used to represent the way strategies are integrated and how they change during learning walks.

Behavioural studies have shown that learning walks involve a progressive increase in movement range as ants accumulate experience [16,22]. Furthermore, ants with different levels of experience exhibit distinct behaviours in the same environment [8], which could indicate that ants form an internal estimate of environmental familiarity that influences their navigational choices [56,57]. Based on these findings, our model introduces the Learning Vector to capture how environmental familiarity shapes the balance between homing and exploration. A testable prediction from this model is that if environmental familiarity is suddenly reduced—for example, by rotating the panoramic visual cues around the nest—ants should shift their exploratory direction inward, favouring homing over further exploration. This effect should be more pronounced in experienced ants, who rely more on visual familiarity, than in naive individuals. Importantly, such experience-dependent changes in directional bias could serve as behavioural evidence for the existence of an internal Learning Vector. By quantifying the angular deviation before and after visual disruption across different experience levels, it may be possible to infer not only the presence of vector-based integration, but also how its parameters—such as cue weights—change with experience. Beyond visual familiarity, our model also suggests two further testable predictions about how ants determine their primary exploration direction during learning walks. First, in our supplementary experiments we explored learning walks under asymmetric olfactory fields using the model. Within the Learning Vector framework, the transition from no olfactory input to odour-guided navigation suggests an increased exploratory tendency. Accordingly, if the environment contains an asymmetric olfactory field, the primary direction of early learning walks may occur more often on the side with stronger odour intensity. This prediction could be tested behaviourally by creating controlled asymmetry around the nest and observing whether real ants preferentially direct their learning walks toward the higher-concentration side. Second, neuroscience studies [58,59,60] have shown that Kenyon Cell–MBON synapses in the mushroom body are modulated by dopaminergic neurons delivering appetitive or aversive signals [60]. We therefore hypothesize that such valence-based learning mechanisms may also influence exploration directions during learning walks, with rewarding experiences reinforcing fidelity to certain directions and aversive experiences promoting avoidance and diversification. In our model, this mechanism could be further extended by incorporating dopaminergic neurons into the mushroom body to evaluate learning walk experiences, and from a neuroscience perspective, this prediction could be tested by examining whether DANs are activated during learning walks, thereby probing whether ants evaluate their experiences as appetitive or aversive. Our model assumes that the integration of multiple strategy cues is governed by Equation (Equation 11), and this formulation enables the reproduction of key trajectory patterns and learning features observed in learning walks. These results raise the possibility that, at the neural level (see the following paragraph for a more detailed discussion), the PI system interacts with the visual learning system and the innate olfactory system through inhibitory connections.

Biological studies have revealed considerable interspecific and individual variability in the exploration strategies exhibited during learning walks [16,17,22]. Consistent with these observations, our model reproduces substantial diversity at the individual level including differences in simulated trajectories, variations in the effectiveness of visual learning, and the dynamic weighting during the course of learning walks, as revealed through trajectory simulations and quantitative analyses of cue weighting dynamics. In addition, by manipulating the degree of weight dependence on exploratory experience, we were able to simulate population-level differences, providing a framework to investigate how behavioural diversity emerges across individuals and species. The diversity of exploratory behaviours observed in our model prompted us to investigate the underlying sources that contribute to this variability. Based on insights from our simulations and existing biological evidence, we hypothesized that three main factors may contribute to this diversity: intrinsic differences in neural architecture, variability in environmental information structure, and experience-dependent agent–environment interactions.

Among the factors contributing to behavioural diversity during learning walks, intrinsic variability in neural architecture is likely to play an important role. In our learning walk model, we adopted a Mushroom Body (MB) framework [48] that reflects anatomical findings of substantial inter-individual variability in Kenyon Cell (KC) connectivity [61,62,63]. Following this constraint, different agents in our simulation exhibit stochastic wiring between Visual Projection Neurons (VPNs) and KCs, introducing differences in how rapidly and efficiently individuals learn and associate environmental views. These internal differences ultimately lead to variations in cue weighting patterns and diverse exploratory trajectories, consistent with the individual variability observed in empirical studies. In addition to intrinsic factors, variabilities in the structure of environmental information also substantially influences exploratory behaviours. In visually rich environments, abundant and reliable landmarks enable individuals to acquire detailed spatial memories and rely more heavily on visual navigation [16,17,22], whereas in visually sparse habitats, the limited availability of salient features may encourage greater reliance on path integration or other non-visual modalities during early learning [6,17,55]. Such environmental differences can drive divergent navigational strategies both within and across species, further contributing to behavioural diversity. Finally, beyond these relatively static sources, biological interactions with the environment dynamically shape behavioural diversities. The specific directions chosen during early exploration influence the spatial information encountered, shaping the memories formed and subsequently biasing future decision-making processes [8,21,22]. Thus, early exploratory experience modulates subsequent learning strategies and reinforces individual differences over time.

To explore the potential mechanisms underlying multimodal integration during learning walks, we adopted a vector fusion [23,27,30,31] approach that offers a simple yet biologically constrained framework for modelling behaviour. This strategy provides an initial step toward understanding how insects may dynamically combine multiple navigation cues in a manner consistent with known biological principles. As a further development of the current model, implementing the proposed cue fusion mechanism at the neural circuit level would represent a valuable next step. Anatomical and functional studies indicate that the Mushroom Body (MB), the Lateral Horn (LH), and the Central Complex (CX) play complementary roles in insect navigation. The MB is associated with learned valence and associative memory [63], particularly in the context of visual [29] and olfactory cues [61]. In contrast, the LH is implicated in innate olfactory-driven behaviours and rapid sensorimotor responses [64]. The CX is thought to support spatial orientation, path integration, and vector computations [65,66]. Given these anatomical links, it is therefore plausible that the proposed Learning Vector fusion mechanism may be implemented within the CX, where inputs from the MB and LH—carrying associative and innate cue information—can be integrated with direct path integration signals [31,67,68]. We hypothesise that the MB, LH, and the internal compass within the CX are integrated within a ring attractor network [49], where environmental familiarity may shift the attractor state and move the homing vector into a learning vector (desired heading) that guides early learning walks. Once such a desired heading is formed, the fan-shaped body can compare it with the current heading, and asymmetric activity in the lateral accessory lobes (LALs) may bias motor output, thereby steering the insect toward the desired direction [66]. This hypothesis provides a biologically grounded basis for future work aiming to uncover how spatial decisions emerge from converging multimodal inputs within the insect brain.

Each strategy module in our model was implemented in a simplified form, reflecting a practical modelling decision aimed at capturing the overall behavioural dynamics of learning walks rather than the fine-scale biological details of individual subsystems. As a first step toward modelling multisensory integration during learning walks, we adopted a NaviNet-style architecture [69] (in which largely independent sensorimotor modules, such as vision and path integration, are coordinated by a network of motivation units that encode behavioural states and switch between them according to internal or external cues) in which each sensory subsystem was implemented in a minimal yet functional form. While simplified, these components collectively establish a foundational structure for future models aiming to incorporate greater biological detail and mechanistic realism. For the visual subsystem, previous work has proposed compass-modulated encoding mechanisms that enhance the robustness of spatial representations and improve biological plausibility [39]. For the path integration module, more mature neural models have been developed [27,65], and extensive studies have also explored its interaction with visual navigation systems [31,38,40,66]. For the olfactory subsystem, incorporating wind direction and plume structure into modelling would better reflect the ethological reality of odour-guided behaviour [70]. These directions represent promising avenues for future refinement of each sensory module in both neural and behavioural modelling contexts.

It has been observed in the field that ants undertake different times of learning walks before their first foraging walk [16,21,22,52,55]. In biological systems, the decision to conclude early learning is likely regulated by internal mechanisms [14,21,52], potentially based on accumulated familiarity with the environment or confidence in the acquired spatial representation. However, it is still unclear how the transition of the animal’s behavioural state from learning to foraging happens. In the proposed model, the termination of the learning walk phase is manually imposed for simplicity. But incorporating such a biologically plausible stopping criterion—such as a dynamic threshold on environmental novelty—may enhance the autonomy and ecological validity of the model.

Our model of insect learning walks, which combines a “learning vector” with dynamic weighting across multiple sensory and navigational subsystems, offers several directions for robotic navigation. One clear application lies in loop closure for SLAM [71]. Here, a robot could use biologically inspired checks of familiarity—such as matching visual scenes, odour-like cues, or path integration signals derived from gyroscopes and accelerometers—to trigger loop closure events more reliably. Our model also reflects the balance between exploration and safety. In robotics, this points to an exploration strategy where, at the start of deployment, a robot performs short, learning-walk-like movements—scanning, turning, and circling its base—to build up “local familiarity fields” before travelling further. The decision to explore or return can then emerge from a weighted integration of sensory memory and exploratory drive. In addition, bootstrapping in unfamiliar environments could be achieved through these same initial routines, providing robots with a first set of sensory snapshots (visual, chemical, inertial) before full exploration, echoing scaffolded or curriculum learning approaches in robotics [72]. Finally, because insect navigation achieves remarkable efficiency with minimal resources, our model highlights how similar trade-offs could enable lightweight deployment on small, low-power robots [73]. Taken together, these principles show how insect-inspired navigation can inform robotic design, supporting robustness while also shedding light on how sensorimotor learning links neural control, body dynamics, and the environment [42,43,44,45]. 

## Figures and Tables

**Figure 1 biomimetics-10-00736-f001:**
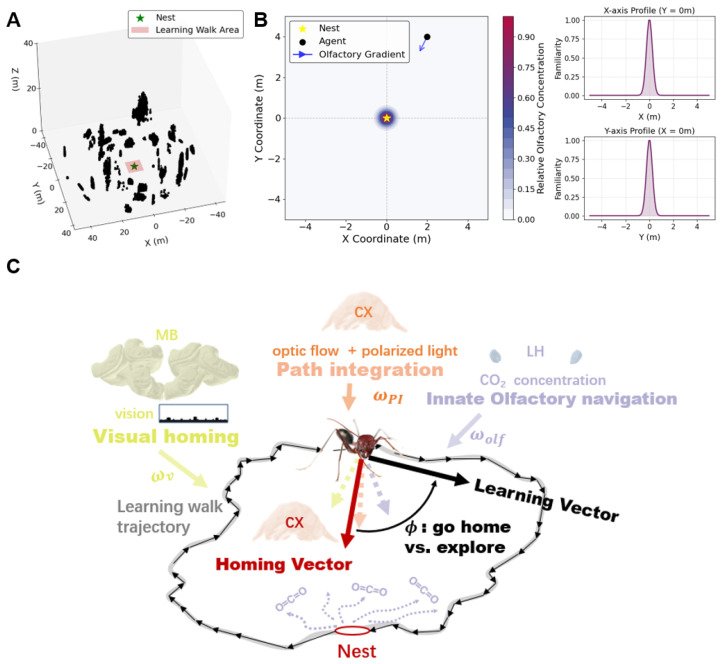
Multimodal sensory environment and model architecture for learning vector formation. (**A**) Three-dimensional rendering of the simulated terrain used during the learning walks. The nest is indicated by a green star, and the designated learning area is highlighted. (**B**) Top view of the olfactory gradient centred on the nest. The agent’s position and the local gradient direction are indicated. The right panel shows cross-sectional profiles of the odour concentration along the X and Y axes, illustrating the Gaussian symmetry of the distribution. (**C**) Schematic of the multimodal integration model. Involved brain regions include the central complex (CX) for path integration, the mushroom body (MB) for visual input, and the lateral horn (LH) for olfactory cues. These regions provide directional and familiarity-related signals that are ultimately integrated within the CX into a Homing Vector, which is further modulated by angular offset to generate the Learning Vector used for navigation decisions.The brain region illustration is adapted from https://www.insectbraindb.org/app/ (accessed on 30 April 2025), and the ant image is adapted from https://www.shutterstock.com/zh/image-photo/desert-ant-cataglyphis-bicolor-isolated-on-2163663957 (accessed on 30 April 2025).

**Figure 2 biomimetics-10-00736-f002:**
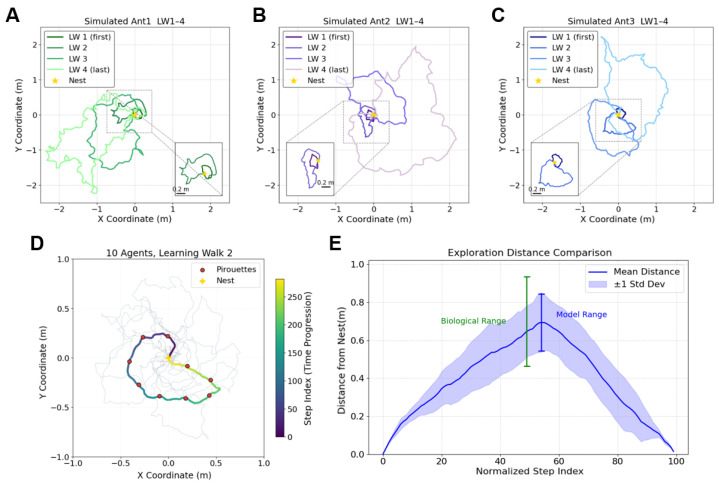
Learning walk trajectories and exploration analysis in simulated ants. (**A**–**C**) Trajectories of Simulated Ants (Ant1, Ant2, and Ant3) during learning walks (LW1–4). The darkest colour represents the first learning walk (LW1), with progressively lighter colours indicating subsequent walks (LW2, LW3, and LW4). The yellow star marks the nest location in each panel, and the inset shows a zoomed-in view around the nest (0.2 m). Each panel corresponds to a different simulated ant, illustrating the progression of their exploration behaviour over multiple learning walks. (**D**) Trajectories of 10 agents during learning walk 2, with red circles indicating pirouette events and yellow stars marking the nest location. The colour bar represents the step index (time progression) during the walk. Shaded grey trajectories represent the paths of all ten agents, demonstrating variability in coverage and progression. (**E**) Exploration distance from the nest plotted over normalized step indices. The blue curve shows the mean distance with the shaded region representing ±1 standard deviation across individuals. The green vertical bar marks the biological exploration range observed in empirical studies [22], who reported an average exploration distance of 0.184±0.0636 m during the first learning walk, with the mean distance increasing by a factor of 3.8 in the second walk. As the original study did not provide variability for the second walk, we conservatively assumed that the standard deviation scaled proportionally (3.8×), consistent with their figures showing increased variability. The blue shaded region represents the model-predicted exploration range, highlighting the alignment between simulated and biological data.

**Figure 3 biomimetics-10-00736-f003:**
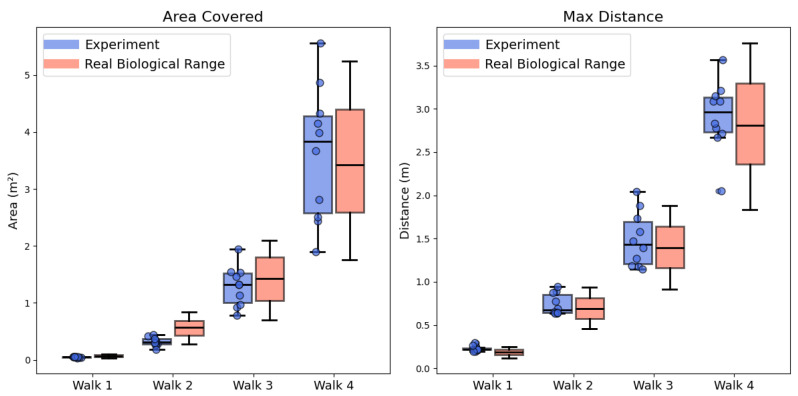
Comparison of simulated and biological data for exploration area and maximum distance across learning walks. **Left panel**: Boxplots showing the exploration area covered by ten simulated ants over four learning walks, compared to empirically reported biological ranges (red boxes). The biological data are derived from [22], who reported that the area covered during the first learning walk was $0.07\pm 0.035\;m^{2}$ (mean ± s.d.). For the subsequent three learning walks, the mean values were reported to increase by factors of 8-fold, 2.5-fold, and 2.5-fold, respectively. As [22] did not report how the standard deviation changed across walks, we conservatively estimated that the standard deviations scaled proportionally with the corresponding mean increases to obtain the biologically constrained ranges. **Right panel**: Boxplots illustrating the maximum distance from the nest reached by simulated ants, again compared with empirical biological data. The maximum distance reached was reported as 18.4±6.36cm (mean ± s.d.), with the mean increasing by factors of 3.8-fold, 2-fold, and 2-fold in the subsequent three learning walks. As [22] did not provide information on how the standard deviation changed, we conservatively assumed that the standard deviations scaled proportionally with the corresponding mean increases to obtain the biologically constrained ranges. Across 40 trials, 85% of the simulated values for exploration area and 90% for maximum distance fell within the corresponding biological ranges, demonstrating that the model accurately captures both the scale and variability of real ant exploratory behaviour.

**Figure 4 biomimetics-10-00736-f004:**
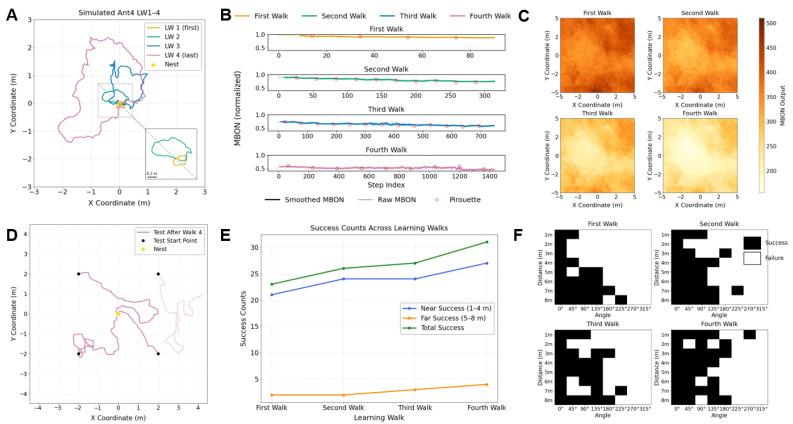
Learning walk modelling and homing performance analysis. (**A**) Trajectories of a single simulated ant across four sequential learning walks (LW1–4). Each learning walk is shown in a different colour: LW1 (first) in orange, LW2 in green, LW3 in blue, and LW4 (last) in purple. The yellow star marks the nest location, and the inset provides a zoomed-in view around the nest (0.2 m). (**B**) Real-time MBON responses recorded during the four learning walks. Raw MBON values are shown as light lines, with corresponding smoothed MBON traces overlaid in dark colours. Pirouette events, indicating moments of abrupt turning associated with view learning, are marked along the trajectories. (**C**) Heatmaps of simulated MBON output across the learning area after each of the four learning walks. In the model, simulated MBON activity decreases uniformly across the entire tested space as learning progresses. The Appendix A further presents normalized simulated MBON activity for each individual learning walk to highlight relative changes within the learning area. (**D**) Homing test trajectories after the fourth learning walk, starting from four distinct test points. Successful trajectories are shown with high opacity, whereas unsuccessful trajectories are rendered with reduced transparency. (**E**) Success counts across successive learning walks. Lines represent total success (green), near success (1–4 m, blue curve, largely within the learning walk area), and far success (5–8 m, orange curve largely outside the learning walk area) based on final homing distance. (**F**) 2D representation of success and failure outcomes at different test points. Black squares indicate successful homing, and white squares indicate failures, for each learning walk.

**Figure 5 biomimetics-10-00736-f005:**
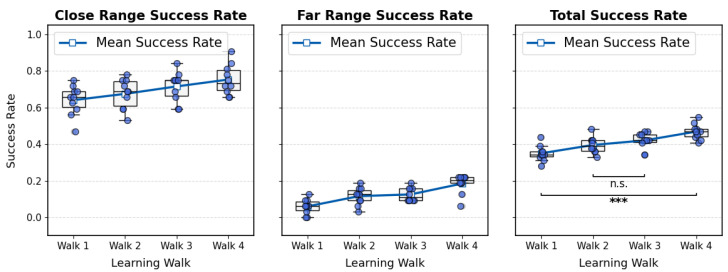
Homing success rates across four learning walks. Success rates of ten agents during the homing ability test across four learning sessions. The three panels from left to right represent: (1) Close-range success rates (1–4 m, normalized over 32 test points); (2) Far-range success rates (5–8 m, normalized over 32 test points); and (3) Overall success rates across all 64 test points. All three plots show a consistent increase in homing success with additional learning walks. *** Statistical annotations indicate significant improvement from Walk 1 to Walk 4 (*t*-test, *t* = 7.740, *p* = 0.00003, $p_{\mathrm{holm}}$ = 0.00018), and n.s. no significant difference between Walk 2 and Walk 3 (*t*-test, *t* = 1.121, *p* = 0.29135, $p_{\mathrm{holm}}$ = 0.29135).

**Figure 6 biomimetics-10-00736-f006:**
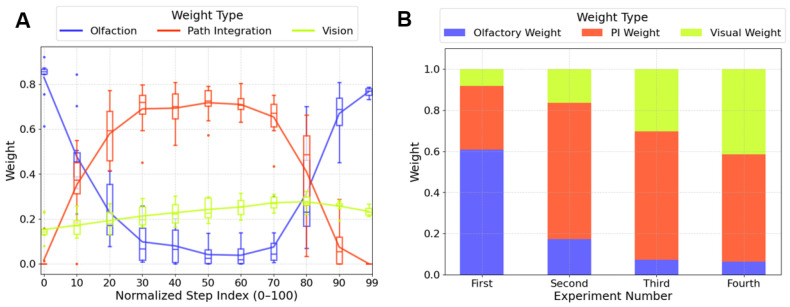
Dynamic weighting mechanism of olfactory, path integration, and visual cues during learning walks. (**A**) Cue weight dynamics throughout the second learning walk, generated by our proposed mechanism and illustrated using ten simulated ants. Boxplots represent the interquartile range of weights across individuals at each normalized step, showing reproducible patterns in the model. In the simulations, olfactory weight (blue) is initially dominant but decreases with distance from the nest, path integration weight (red) peaks during mid-range movement, and visual cue weight (yellow-green) gradually increases. (**B**) Average cue weights at the end of each learning walk. The proposed mechanism produces a progressive shift from early reliance on olfactory and path integration cues to vision-dominated strategies in later walks, a trend that aligns with behavioural findings reported in real ants [11,21].

**Figure 7 biomimetics-10-00736-f007:**
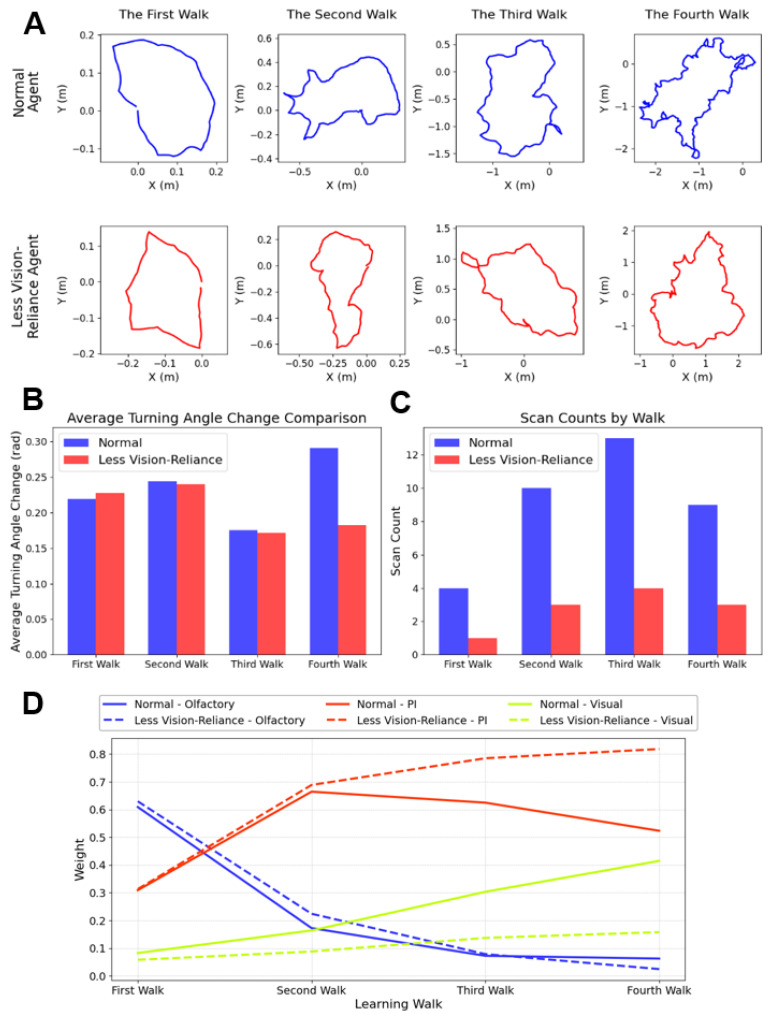
Comparative analysis of learning walk behaviour under normal and reduced visual weight conditions. (**A**) Representative learning walk trajectories across four sessions. Upper panels: normal agents (blue); Lower panels: reduced visual reliance agents (red). Trajectories in the reduced vision condition showed more localized and irregular patterns, suggesting altered exploration behaviour. (**B**) Average turning angle change per step across learning walks. Agents under the normal visual condition exhibited smoother turning behaviour. (**C**) Number of scan events (pirouette-like turns) generated per learning walk in the simulations. Agents under the normal condition produced more scans. (**D**) Average cue weights across learning walks for normal agents (solid lines) and agents with reduced visual reliance (dashed lines). Separate trajectories represent the weighting of olfactory, path integration, and visual cues. In the model, reducing visual reliance resulted in increased use of path integration and olfactory cues, illustrating the flexibility of the proposed multisensory weighting mechanism.

**Table 1 biomimetics-10-00736-t001:** Summary of model parameters used in the Learning Vector framework. Static parameters have fixed values; adaptive parameters are updated based on previous exploration performance or internal states.

Parameter	Description	Value/Setting
$C_{0}$	Normalised olfactory concentration at the nest	1
σ	Spread of the Gaussian odour field	0.15–0.25
*U*	Step size per simulation step	0.01 m
$\lambda_{\mathrm{vis}}$	Visual learning rate in MB network	0.01–0.018
$\theta_{\mathrm{KC}}$	Kenyon Cell activation threshold	0.04
$N_{\mathrm{VPN}}$	Number of Visual Projection Neurons	81
$N_{\mathrm{KC}}$	Number of Kenyon Cells	4000
δ(t)	Path integration noise per step (Gaussian)	N(0,0.0001)
$F_{v},F_{\mathrm{olf}},F_{\mathrm{PI}}$	Familiarity-to-offset mapping functions	Piecewise linear
$n_{p}$	Step limit per learning walk	Adaptive
$I_{\mathrm{scan}}$	Visual scanning interval	Adaptive per walk
*r*	Step scaling factor during learning walks	2.0–2.5
$p_{\mathrm{rand}}$	Random step probability	0.2–0.8
κ	von Mises concentration parameter	5–50

**Table 2 biomimetics-10-00736-t002:** Statistical comparison of homing success rates across learning sessions. The table reports test methods, *t*-values and *p*-values based on pairwise comparisons of homing success distributions, aggregated across all 64 release points. While the difference between LW3 and LW2 was not statistically significant, the comparison between LW4 and LW1 shows a highly significant improvement, indicating cumulative effects of repeated learning walks on visual navigation performance.

Comparison	Method	*t*	*p*	Adjusted *p* (Holm)
LW2 vs. LW1	*t*-test	3.783	0.00433	0.01299
LW3 vs. LW2	*t*-test	1.121	0.29135	0.29135
LW4 vs. LW3	*t*-test	4.147	0.00250	0.01000
LW3 vs. LW1	*t*-test	5.352	0.00196	0.00980
LW4 vs. LW2	*t*-test	4.045	0.00591	0.01299
LW4 vs. LW1	*t*-test	7.740	0.00003	0.00018

**Table 3 biomimetics-10-00736-t003:** Effect sizes (Cohen’s *d*) for homing success rates across learning sessions. Pairwise effect sizes are reported for comparisons of homing success between different learning walks. Large effect sizes (e.g., LW4 vs. LW1) indicate substantial improvements in performance after repeated learning walks, whereas smaller values (e.g., LW3 vs. LW2) suggest more modest gains.

Comparison	Cohen’s *d*
LW2 vs. LW1	1.037
LW3 vs. LW1	1.595
LW4 vs. LW1	2.799
LW3 vs. LW2	0.553
LW4 vs. LW2	1.684
LW4 vs. LW3	1.100

## Data Availability

The original contributions presented in this study are included in the article/Appendix A. Further inquiries can be directed to the corresponding author.

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
