# Peer review of "A Vector-Based Computational Model of Multimodal Insect Learning Walks"

_biomimetics, 2025, doi:10.3390/biomimetics10110736_

Round 1
Reviewer 1 Report
Comments and Suggestions for Authors
The manuscript “A Vector-Based Computational Model of Multimodal Insect Learning Walks” by Xiang et al. addresses an exciting topic in neuroethology: how do insects integrate multimodal information for adaptive behavior? The authors propose a computational model to recapitulate how ants use various information classes (vision, olfaction, path integration system) during learning walks. Specifically, they suggest that the different information classes are fused into what they call a “learning vector” that deviates from the homing vector in such a way that the degree of deviation reflects unfamiliarity (a large angle between the homing vector and the learning vector = more familiarity, resulting in more exploration).
However, I have concerns regarding the integrity of the manuscript, as many aspects of it strongly resemble common suggestions made by AI (e.g., ChatGPT typically suggests that biological studies offer potential insights for bio-inspired robotics). This, in turn, results in a combination of unspecific, supposedly “impactful” wording that does not directly reflect the results, and conclusions that appear distant from the run simulations. Moreover, the core of the model has already been published (https://doi.org/10.7554/eLife.54026; one of the authors of the present manuscript, Xuelong Sun, was the first author of the previous publication), and the present manuscript fails to fully disclose this strong dependence on previously published code. To be fair, the authors cite their previous work (Sun et al. (2020); citation 26) in the present methods section, but reusing its core functions and corresponding text should be indicated more clearly. As a reader, “using an associative learning rule […] following approaches established in previous models” (l. 167-168) does not indicate a complete reuse of previous work. Given the strong dependence on previous work, there is a strong overlap in wording (e.g., “2.1.1 Visual Environment” in the present manuscript is essentially the same as the M&M subsection “Simulated 3D world” in Sun et al. (2020); consequently, I assume the same environment was used here). Consequently, many aspects that were criticized in Sun et al. (2020), are also applicable here:
- Lack of explanation of what Zernike Moments are
- A clear separation between model assumptions and biological facts
- Demonstration of model robustness through, e.g., obstacles or changing the complexity of the virtual environment
Lastly, I believe that the code, as it is stored on GitHub, cannot be run because the files “RouteMemory_Test.mat” and “HomeMemory_X0Y0.mat” are missing from the repository (I did not attempt to run the code myself). Many comments, supposedly documenting the code, are not in English, which would make it difficult to reproduce the reported results and use it as a general tool.
--- SPECIFIC COMMENTS ---
ABSTRACT
According to https://scispace.com/ai-detector, 100% of the Abstract resembles text produced by AI. Although tools like ChatGPT are great for non-native speakers to improve their writing, incorporating common AI arguments, such as proposing that biological results can be applicable to robotics, should at least be acknowledged.
Throughout the text, but not consistently (e.g., see l. 54-55), the authors capitalize “learning walks”. I would stick to what is commonly done in literature: not capitalizing it.
Throughout the text, the authors treat “path integration” as a sensory modality. However, it is already the result of integrating various information classes.
INTRODUCTION
- The introduction would benefit from clear statements/definitions of navigation vs orientation.
- Cite which ethologists (l. 41-42) and which neuroscientists (l. 45-46).
- (l. 47-48) be more precise with which neuroanatomical structures and which changes
- (l. 48-49) To what extent are learning walks important, and what happens when an ant returns to its nest, but the environment is a different one?
- (l. 51-55) cite how learning walks impact these fields and which studies provided valuable insights
- (l. 58) “neuroplastic changes driven by multisensory experiences” is an extremely vague statement, and I don’t see how this leads to the consequence that learning walks require complex environments.
- (l. 67-68) There are many studies and models on multimodal cue integration. The introduction (or later discussion) could have addressed some of them. For example, Arganda et al. (2012) https://doi.org/10.1073/pnas.1210664109 propose a decision-making rule based on the Bayesian integration of different information classes, taking into account the reliability of each class. Such Bayesian principles are commonly employed and should be mentioned and discussed. Also, consider Cheng et al. (2007) https://doi.org/10.1037/0033-2909.133.4.625. On a similar note, what happens if different information classes are in conflict in the proposed model?
- (l. 77-78) vector-based modelling approaches are not new and are commonly used, so how does this render the used model “novel”
- (l. 81-85) This part represents a strong simplification of complex neural circuits, and the authors should explain why the olfactory aspect of the model is restricted to the LH and does not involve the MB. Does it even need a link to presumable neural correlates, where the authors assume the information to be class-specific directional information to be present?
- (l. 93) explain which biological hypotheses you are referring to
- (l. 96) Provide more information about which learning walk patterns you are referring to
Overall, the introduction should be more concrete in specifying the aspects it refers to. The authors should clearly state what navigation is and what orientation is; the behavior of learning walks, which ants do it when/why, and which sensory modalities are present; which sensors are involved in accumulating unimodal evidence and how it is processed; how different modalities (e.g., vision and olfaction) result in directional information and how/where it is integrated; what path integration is. Together, this should help precisely identify the new aspect of the manuscript (as explored during learning walks).
METHODS
- The authors should explain the reasoning behind parameter selection. For example, the modeled odor concentration gradient drops rapidly and is effectively zero after ±1 distance units (cm, I guess?). From which distance can real ants locate their nest using olfactory information only?
- (l. 124-125) Why is it sufficient, and on what data do you base this claim?
- (l. 136-137) The authors repeatedly refer to different navigation strategies, but fail to properly introduce them. Generally, this section (2.2.) would benefit from more details and especially an illustration explaining the different aspects.
- (l. 138-139) The authors could explain more of the behavioral principles behind their model
- (l. 163-169) As mentioned above, the underlying principles of the model are somewhat abstract. Especially the section about the processing of visual information would benefit from an illustration and a more accessible description. Where exactly are Zernike Moment amplitudes processed and what are exemplary inputs and corresponding outputs at each stage of the “Visual Learning”-module? None of the equations contains Ai(θ).
- (l. 180) The authors use a complex approach for the processing of visual information, but strongly simplify the olfactory environment and processing of derived information. Can ants determine the concentration gradient of olfactory information as “easily” as described?
- (l. 195-196) The authors try to base their PI module on neural circuits in the brain and claim their approach to PI is a neural model constrained by biological facts. However, what follows is a simple vector summation. I do not see how this resembles a neural network.
- (l. 202) What kind of noise and with which strength was used?
- (l. 212) I would not label PI as a sensory modality.
- (l. 212-227) Instead of recapitulating their selling arguments from the abstract, I was expecting a clearer description of the actual methods.
- (l. 228-231) Given accumulated noise in the PI module, I would assume that PI reliability decreases over time. Given that the authors try to model a learning vector based on the continual integration of different information classes, I would be interested in how Bayesian integration, which could incorporate both signal strength and reliability for each information class, performs in comparison to the proposed approach.
- (l. 238) This reciprocal dependence of PI reliability and the reliability of the other two information classes will result in flawed arguments in later parts (Fig.6) where the authors “find” that different information classes are dynamically weighted. However, all they show is that their formula 11 works (large wPI when the other two weights, or one of them, are small – and vice versa).
- (l. 254-255) again, I don’t think that PI represents a sensory modality (I will not further comment this below anymore)
- (l. 267-270) How exactly is familiarity calculated? The weights for vision and olfaction are in the range of [0, 1], but the PI vector can exceed 1. How is this taken into account?
- (l. 294) With which cutoff threshold are scans initiated?
- (l. 303-304) How is the agent initialized, and how does it avoid the agent immediately turning back to the nest as everything is unfamiliar during its first steps out of the simulated nest?
- (l. 316) The authors claim to replicate biological data. Which species?
- (l. 320) How does a von Mises distribution with the chosen parameters look like? How were the distribution’s parameters chosen?
- (l. 326) How were the parameters chosen? Also, some parameter letters, such a “r” and “C”, are often reused as variables.
- (l. 330) Stating that an insect's ultimate goal of learning is homing is a bold argument. I’d turned it down.
- (l. 331-332) How would simulated experiments under uncontrolled conditions look?
- (l. 332) Why only simulate 10 agents?
- (l. 342-347) This reads like it was written by AI. Also, no effect sizes were reported.
- (l. 348) Instead of changing the agents’ visually guided behavior, I would use species-specific environments and scanning frequencies to enable an analysis of interspecific variability. Alternatively, which two species live in the same habitat but differ in their visual learning capabilities? How do the authors justify the way the parameters were changed?
RESULTS (figures are addressed separately)
- (l. 359-360) Pleas explain how the experimental settings are similar and exactly which biological data were extracted from which study and how they relate to your model.
- (l. 382-383) Based on how the authors implemented the learning walk direction, random exploration in one direction during the first learning walk should promote stronger exploration in that direction, right? Actually, a deeper analysis and description of what can be seen and why would be desirable.
- (l. 391) I have trouble finding the corresponding values in the referenced experimental study [21].
- (l. 401) Given my criticism of the figures below, I do not support the authors’ claim that their model can form the computational basis of ant learning walks
- (l. 413 and following) Without a description of the underlying model, I have trouble understanding how “MBON activity” can be investigated. Moreover, the authors should further highlight that no MBON activity was directly measured but simulated with a neural network.
- (l. 435-438) Do the data allow for using a t-test? Why did the authors additionally pool the data?
- (l. 445-446) I do not have access to the supplementary figures
- (l. 453-454) What do the authors mean with “internal and behavioral perspectives”
- (l. 455-459) This observation could also be due to problems with the model’s implementation and not with the underlying biology.
- (Tbl. 2) This table was not referenced in the text, and the test design would require compensation for a multiple comparison.
- (l. 475-498) The reason why olfactory cues dominate the model at the beginning and at the end of each learning walk is due to the model’s implementation and the agent’s proximity to the nest in those phases. Consequently, this affects the PI weight simply due to the implementation of those weights. From this, I would not derive that real ants adjust cue waits context-dependently (even if they probably do so).
- (l. 503-505) To which other species do the authors compare Cataglyphis?
- (l. 509-510) Here, the authors acknowledge that different species experience different visual habitats. So why did they not change habitat structure?
- (l. 529-530) The authors never clearly stated based on which species their model was based on. Moreover, any model with appropriate parameters and architecture is capable of replicating trajectories that resemble those produced by real animals.
FIGURES
- (1A) Axes do not have units, and the aspect ratio is not equal
- (1A) Learning walk area is mostly occluded by the visual features
- (1B) Axes do not have units. If in cm, do learning walks only occur within +/- 5cm around the nest entrance? As mentioned above, I am concerned about the highly localized olfactory gradient
- (1C) Sources for brain region illustrations and ant cartoon are not provided
- (2A-D) Axes do not have units. If in cm, do learning walks only occur within +/- 3cm around the nest entrance?
- (2E) y-Axis lacks units. A source for biological data was not provided.
- (3) Source for biological data was not provided. I checked Deeti & Cheng (2021; https://doi.org/10.1242/jeb.242177), that was mentioned in the text, and their learning walk characteristics do not fit (e.g., areas ranging from 0.1 to 10m² Deeti & Cheng (2021) in comparison to values not exceeding 5m² in the present study) . Show all data points together with boxplots.
- (4) No panel labels appear in the figure, despite being referenced in the legend.
- (4-top left) Axes do not have units. If in cm, do learning walks only occur within +/- 3cm around the nest entrance? Description of color coding does not match the figure
- (4-top center) As mentioned earlier, I do not understand what the authors mean with MBON
- (4-top right) The authors write that there are important differences between the 4familiarity maps. However, stacking the maps does not allow for spotting differences. 2D heatmaps and quantifying differences between them would be a better approach. Label of the z-axis is partly occluded.
- (4-bottom left) Axes do not have units. If in cm, do learning walks only occur within +/- 5cm around the nest entrance?
- (4-bottom center) What is the rationale behind pooling near and far success? Success rate as shown in Figure 5-left and 5-center should be sufficient
- (4-bottom right) Axes labels are almost too small to red. Is this data for one agent or across many? Why not show success rates across all agents as a heatmap? “Spatial distribution” is a misleading term here. It is simply a 2D representation of the data.
- (5) Success for close-range homing is always high and always low for far-range trials. Why is the total success rate not bimodal, given it relies on the close- and far-range data? Consider adding an intermediate-range condition. Show individual datapoints along with boxplots. Still, I do not know why pooling across both conditions was necessary? Maybe to get statistical significance? Instead of reporting asterisks, include the p-value as well (or at least explain what *** means). Add tests for clos- and far-range only data.
- (6) As mentioned above, I believe that the “dynamic weighing” is rather an artifact of the model’s implementation rather than a finding.
- (7A) Axes do not have units. If in cm, do learning walks only occur within +/- 1cm around the nest entrance? How come these values are even smaller than before?
- (7B) y-axis does not have units.
- (7C) The strong reduction in scan count is rather an artifact of the model’s implementation (scan probability reduced to 1/4) rather than a finding.
- (7D) The strong reduction in visual reliance is due to the implementation of the model and not a finding.
CONCLUSION AND DISCUSSION
- According to https://scispace.com/ai-detector, 80% of this section resembles text produced by AI. Thus, most arguments appear to make promises without addressing measurable results, and the section reads like “AI-paperese.”
- Sudden changes in environmental familiarity (e.g., 558-559) and variability in environmental information (l. 594-595) are discussed but not tested. However, testing this should be easy given the modularity of the world.
- How do the authors explain apparent differences in tortuosity between their data (e.g. Fig.2D and that observed in real animals (e.g., Deeti & Cheng (2021) Fig.3)
- (l. 569-570) The discussed inhibitory connections come a bit out of nowhere. The authors could discuss in detail the involved neural substrates for all 3 information classes and where they are processed, integrated, and modulated.
- (l. 574) Which “substantial diversity at the individual level” are you referring to? Would this still emerge if all agents started with the same heading direction?
- (l. 622-624) Across species, there is already ample work on the mechanisms underlying spatial decision-making and vectorial representation of goals. Moreover, the CX consists of several neuropils. The authors should expand their discussion and provide concrete hypotheses of how converging multimodal input converges to drive behavioral output.
- (l. 629) What is a NaviNet?
- (l. 645-646) The present study did not involve foraging trips, but only learning and homing. Also, in this context, it would be “behavioral state”.
- (l. 650-665) typical AI closing statement, as criticized above.
Author Response
Thank you for your review. Please read the first part of the attached file LearningWalksResponse1.pdf.

Reviewer 2 Report
Comments and Suggestions for Authors
- What is the main contribution of this study?
- Explain in Eq 11, the weight of the path integration, why this formula was chosen.
- what is the difference between ant colony optimizer (ACO) and your proposed method? Equation 15 and 16 is similar with this algorithm.
- Why is the sensory environment so simplified, and how would results change in realistic settings?
- How exactly could this model be applied in robotics, for example in SLAM or energy-efficient exploration?
- Strongly recommend to review all paper in terms of typos.
Author Response
Thank you for your review. Please read the second part of the attached file LearningWalksResponse1.pdf.

Round 2
Reviewer 1 Report
Comments and Suggestions for Authors
The authors have addressed my concerns, but not fully.
First, the authors should carefully review their manuscript to refer to modeled “ants” as agents only (or similar terminology, but not simply as “ants”) to avoid confusion with real ants.
Second, the authors write that ants are “following the gradient of these chemical cues [9,42, 47]”. The authors should state based on which concrete biological finding, they chose the respective parameter.
Next, I think the addition of an empty environment is great, but variation in environmental complexity is still necessary to further validate the model's capabilities (i.e., changes in the olfactory gradient’s symmetry, such as simulating constant wind; gradually changing the complexity of the visual scene). This is especially important as the other reviewer asked about this as well.
Further, I still believe that exploring the origin of why some agents display preferred exploration directions, while others display a broader range of exploratory directions, will be a valuable addition to go beyond merely recapitulating real ants’ behaviors, offering testable predictions about the underlying neural processes (which is within the scope of the study as it aims to recapitulate underlying circuits within the model’s architecture).
Regarding statistics, a t-test requires that the data in each group be normally distributed and that the variances between groups be equal. The authors should check whether these assumptions are met. Further, “Table 2: Paired t-test results for close-, middle-, and far-range comparisons” requires correction for multiple comparison as well.
Regarding the figures:
- The z-axis in 1A still does not have the same scale as the x- and y-axis.
- 5 in Deeti et al. shows data points for Area Covered and Max Distance. For a valid comparison in the author’s Fig. 3, the authors should include the actual data, rather than or in addition to their extrapolation.
- In Fig.4C → SI MBON2.png: The diverging colormap is inappropriate for sequential data. Furthermore, the chosen color range does not align with what is shown in Fig. 4C (e.g., Fig. 4C suggests that most data should be within [0.3, 1.2], whereas the new SI figures suggest something completely different). Moreover, why does familiarity around the origin decrease over time, whereas it appears to increase around the edges? Overall, supplementary material should be mentioned in the main text, as well as explained in the SI (e.g., figure caption). Regarding the authors’ claim that “rotation helps to avoid occlusion of the z-axis”, I strongly disagreed, as this is only true for one viewing angle. The authors’ intention to show that modeled MBON activity decreases over time is also visible from Fig.4B. Reworking the figure to include the 4 heatmaps should not be too difficult. It could also reveal whether modeled MBON activity decreases uniformly across the whole tested space or not.
- 6, data are partly occluded by legends.
- 7B&C lack error measures such as standard deviations. The y-label in 7B is cropped. The label of Fig.7 is occluding other parts of the figure.
- The authors should aim for consistency in figure label/axes/legend/etc font size and be more careful in general. Repeatedly spotting mistakes suggests to me that the authors are not meticulous in their science, raising doubts about the manuscript in general.
Author Response
We sincerely thank the reviewer for their time and valuable suggestions. Your comments have been very helpful in improving the quality of our manuscript. The detailed revisions can be found in the attached file LearningWalksResponse2.0.pdf.

Reviewer 2 Report
Comments and Suggestions for Authors
Accept in present form
Author Response
We sincerely thank the reviewer for the encouraging comments and for recommending our manuscript for acceptance in its current form. Your positive assessment and previous suggestions have greatly helped us to improve the clarity and quality of our work.
Round 3
Reviewer 1 Report
Comments and Suggestions for Authors
The authors have addressed all my comments.